# The Anti-Virulence Effect of *Vismia guianensis* against *Candida albicans* and *Candida glabrata*

**DOI:** 10.3390/antibiotics11121834

**Published:** 2022-12-16

**Authors:** Elizangela Pestana Motta, Josivan Regis Farias, Arthur André Castro da Costa, Anderson França da Silva, Alberto Jorge Oliveira Lopes, Maria do Socorro Sousa Cartágenes, Roberto Nicolete, Afonso Gomes Abreu, Elizabeth Soares Fernandes, Flavia Raquel Fernandes Nascimento, Cláudia Quintino da Rocha, Cristina Andrade Monteiro, Rosane Nassar Meireles Guerra

**Affiliations:** 1Laboratório de Imunofisiologia, Departamento de Patologia, Universidade Federal do Maranhão, Avenida dos Portugueses, 1966, Ensino Integrado, Bloco 1, São Luís 65080-805, MA, Brazil; 2Programa de Pós-Graduação em Ciências da Saúde, Centro de Ciências Biológicas e da Saúde, Universidade Federal do Maranhão, Avenida dos Portugueses, 1966, São Luís 65080-805, MA, Brazil; 3Laboratório Experimental de Estudos da Dor, Departamento de Ciências Fisiológicas, Universidade Federal do Maranhão, São Luís 65080-805, MA, Brazil; 4Instituto Federal de Ciências e Educação do Maranhão-Campus Santa Inês, Rua Castelo Branco, 1, Santa Inês 65300-000, MA, Brazil; 5Fiocruz Ceará-Rua São José, S/N-Precabura, Eusébio 61773-270, CE, Brazil; 6Laboratóio de Patogenicidade Microbiana, Programa de Pós-Graduação em Biologia Microbiana, Universidade UNICEUMA, Rua Josué Montelo, 1-Renascença, São Luís 65075-120, MA, Brazil; 7Instituto Pelé Pequeno Príncipe, Av. Silva Jardim, 1632-Água Verde, Curitiba 80250-060, PR, Brazil; 8Programa de Pós-Graduação em Biotecnologia Aplicada à Saúde da Criança e do Adolescente, Faculdades Pequeno Príncipe, Av. Iguaçú, 333-Rebouças, Curitiba 80230-020, PR, Brazil; 9Laboratório de Química de Produtos Naturais, Centro de Ciências Exatas e Tecnológicas, Universidade Federal do Maranhão, São Luís 65080-805, MA, Brazil; 10Departamento de Biologia, Instituto Federal do Maranhão, Avenida Getúlio Vargas, No 4, Monte Castelo, São Luís 65030-005, MA, Brazil

**Keywords:** *Vismia guianensis*, *Candida albicans*, vismione D, *Candida glabrata*, CaCYP51, antifungals, lacre

## Abstract

In folk medicine, *Vismia guianensis* is used to treat skin diseases and mycoses in the Amazon region. We evaluated the anti-*Candida* activity of the hydroalcoholic extract from the leaves of *Vismia guianensis* (EHVG). HPLC-PDA and FIA-ESI-IT-MS^n^ were used to chemically characterize EHVG. The anti-*Candida* activity was determined in vitro by the minimum inhibitory concentrations (MIC) against *Candida glabrata* (ATCC-2001); *Candida albicans* (ATCC-90028, ATCC-14053, and ATCC-SC5314), and *C. albicans* clinical isolates. EHVG effects on adhesion, growth, and biofilm formation were also determined. Molecular docking was used to predict targets for EHVG compounds. The main compounds identified included anthraquinone, vismione D, kaempferol, quercetin, and vitexin. EHVG was fungicidal against all tested strains. *C. albicans* ATCC 14053 and *C. glabrata* ATCC 2001 were the most sensitive strains, as the extract inhibited their virulence factors. In silico analysis indicated that vismione D presented the best antifungal activity, since it was the most effective in inhibiting CaCYP51, and may act as anti-inflammatory and antioxidant agent, according to the online PASS prediction. Overall, the data demonstrate that EHVG has an anti-*Candida* effect by inhibiting virulence factors of the fungi. This activity may be related to its vismione D content, indicating this compound may represent a new perspective for treating diseases caused by *Candida* sp.

## 1. Introduction

Candidemia has become one of the most common invasive *Candida* bloodstream infections. The high global incidence of candidiasis can be explained by the increase in the number of susceptible hosts, such as patients submitted to immunosuppressive treatments, long-term use of broad-spectrum antibiotics, use of catheters and probes, and hematopoietic transplants, amongst others. In addition, fungal resistance to antimicrobials has also increased [1,2].

Candidiasis is the fourth most common cause of nosocomial infections in the world, and *C. albicans* is the most frequent causative agent of medical relevance, due to its prevalence in both healthy hosts and those with underlying conditions or immune impairment. *C. albicans* is one of the most prevalent fungal species in the human microbiota, since it asymptomatically colonizes healthy individuals. This microorganism is, therefore, easily detected in the oral mucosa, gastrointestinal tract, urogenital tract, and skin of humans from birth [3].

Although *C. albicans* is still the leading cause of candidemia, increasing proportions of cases in recent years have been attributed to non-*albicans* species that are often resistant to antifungal drugs, including *Candida glabrata*, *C. parapsilosis*, *C. tropicalis*, and *C. krusei* [4].

The medical impact of *C. albicans* infections usually depends on the pathogen’s ability to form biofilms [5]. Adhesion is the first step of the pathogenic process. It is complex and still not completely understood [6]. Among the known virulence factors, adhesion and subsequent biofilm formation by *Candida* sp. confer to the yeast the ability to persist and grow more easily, thus promoting the persistence of infection [1,7].

Several drugs are used to treat *Candida* infections; however, there has been an increase in the number of *C. albicans* and *C. glabrata* cases that are resistant to the available therapies. The latter is an emerging pathogenic fungus that is resistant to Fluconazole [8]. In this context, plant-derived alternatives are an interesting option for the bioprospecting of new compounds which could be used alone or in combination as antifungal agents [9,10]. Thus, plant extracts and their bioactive molecules might be a promising alternative for the treatment of candidiasis, especially extracts with a high content of phenolic compounds [11].

The medicinal effects of *Vismia guianensis* (Aubl.) Chosy, family Clusiaceae, include its antimicrobial activity, which has been associated especially with orange latex exudate from the branches of this tree. In folk medicine, this species is used to treat wounds, ulcerations, skin diseases, dermatomycoses, and herpes, particularly in the Amazon region [12]. *V. guianensis* leaves contain phenolic compounds such as anthraquinones, flavonoids, xanthones, and benzophenones, with anthraquinones being the most common compounds [13,14,15,16]. The plant also contains some other pharmacologically active compounds, such as vismione D and ferruginin, which possess immunosuppressive [13], antioxidant [17], antibacterial, and antifungal activities [18].

Considering the widespread use of *V. guianensis* by the population and its reported biological effects, this study evaluated its anti-*Candida* activity, as well as its effects on early and mature biofilm formation and adhesion. It also sought to provide data from molecular docking and PASS online prediction to help to guide the identification of the most promising *V. guianensis* compounds with antifungal activity against *Candida* sp, which may also show anti-inflammatory and antioxidant activity.

## 2. Results

### 2.1. Characterization of the Plant Material and Extract

We determined the particle sizes of dried and powdered leaves to characterize the plant material and the yield after extraction.

Figure 1 shows the particle size of the leaves and the product obtained from *V. guianensis*, which was classified as a semi-fine powder, with a large portion being retained in mesh M60 (34.09%), followed by mesh M80 (20.73%).

Table 1 shows the results related to the ash content and the total amount of residue from *V. guianensis*. The different hydromodules used to prepare the hydroethanolic extract of the *V. guianensis* leaves (EHVG) presented similar yields, but different ash content and total residues. The hydromodule 1:10 showed the best results, with 97% organic matter, and, therefore, it was chosen for chemical analysis and experiments.

### 2.2. Chemical Profile of EHVG

To determine the chemical profile of EHVG, different hydromodules were evaluated qualitatively by HPLC-PDA at 254 nm. Figure 2 shows the chromatographic profile of the hydromodule 1:10, which qualitatively exhibited the best extraction profile in terms of peak area.

Figure 3 shows the identification of EHVG chemical compounds by *full-scan* spectrum (hydromodule 1:10).

The fourteen compounds identified by flow injection analysis (FIA-ESI-IT-MS) are depicted in Table 2. It is possible to observe a predominance of anthraquinones, flavonoids, and Vismione D.

Figure 4 shows the structures of the compounds identified in the leaf extract of *V. guianensis* (EHVG), according to the spectrum shown in Figure 3 and the compounds presented in Table 2. Herein, it is important to highlight, the structures of Vismione D, anthraquinone F, and kaempferol, which are important markers of the genus *Vismia*.

### 2.3. EHVG Showed Low Toxicity Using Different Assays

To define the best concentrations of the extract of *Vismia guianensis* (EHVG) for the antifungal assays, its hemolytic and cytotoxic activities were evaluated. The extract showed low toxicity, inducing less than 10% cell death and hemolysis (Figure 5A,B) at concentrations ≤ 5 mg/mL. Cell viability was greater than 80% at concentrations ≤ 10 mg/mL (Appendix A).

### 2.4. Antifungal Activity of EHVG against C. albicans and C. glabrata

EHVG showed antifungal potential and fungicidal activity against all tested strains (Table 3). Against *C. albicans* isolates, EHVG MIC values ranged from 3.125–6.25 mg/mL, MFC from 3.125–12.5 mg/mL, and the CFM/MIC ratio ranged from 1–2. Against *C. glabrata*, the MIC was 3.125 mg/mL, and the MFC was 6.25 mg/mL. MIC values were used as a parameter for selecting two strains for the subsequent assays: *C. albicans* (SC5314) and *C. glabrata* (ATCC 2001), both of which are of clinical interest.

#### 2.4.1. Treatment with EHVG Inhibits the Growth of *C. albicans* and *C. glabrata*

Figure 6 shows the fungicidal activity of EHVG. EHVG inhibited the growth of *C. albicans* and *C. glabrata* when tested at two different concentrations. Figure 6A shows the results for *C. albicans* SC5314, after 36 (2× MIC = 12.5 mg/mL) and 48 h (1× MIC = 6.25 mg/mL). Figure 6B shows similar results against *C. glabrata* (ATCC 2001), in a shorter period, for both concentrations, since the inhibition was observed after 24 and 36 h at concentrations of 2× (6.25 mg/mL) and 1× the MIC (3.125 mg/mL), respectively.

#### 2.4.2. EHVG Inhibits *C. albicans* and *C. glabrata* Adhesion

*C. albicans* (ATCC 90028, SC5314) and *C. glabrata* ATCC 2001 were used to investigate the effects of EHVG on pathogen adhesion. Figure 7A shows that EHVG was more effective than Amphotericin B in inhibiting *C. glabrata* (ATCC 2001) adhesion at the lowest tested concentration (1/2× MIC). At the highest concentration (1× MIC), all treatments had similar inhibitory effects in comparison with the untreated control. Figure 7B,C demonstrate that EHVG effects were equivalent to those of reference drugs (FLU and Ampho B), regardless of the concentration used, against the two different *C. albicans* strains.

#### 2.4.3. EHVG Disrupts Early and Mature Biofilm Formation

Figure 8 shows that treatment with EHVG impaired early *Candida* biofilms formed within 48 h (Figure 8A,C,E) and mature biofilms formed within 72 h (Figure 8B,D,F). This effect was similar to that of Amphotericin B or Fluconazole compared to untreated controls.

Treatment with EHVG inhibited biofilm formation with an efficacy similar to that of Fluconazole or Amphotericin B, even at sub-inhibitory concentrations (1/4 MIC and 1/2 MIC). For *Candida albicans* SC5314, the reduction was more significant than that observed with Amphotericin B (Figure 9).

Figure 10 shows the reduction in biofilm biomass in the groups treated with EHVG for 48 and 72 h, with the same intensity as that observed for Amphotericin B and sometimes higher than that for Fluconazole. This confirms the data obtained from the MTT assay.

### 2.5. In Silico Biological Activity and Toxicity for the Compounds Identified in EHVG

The PASS online tool evaluated the antifungal, anti-inflammatory, and antioxidant activities of the five selected compounds detected in EHVG, including catechin, kaempferol, anthraquinone, and Vismione D, in comparison to those of Fluconazole and Amphotericin. The compounds showed greater Pa than Pi as antifungal, considering Pa values higher than Pi (>0.3) due to the strong molecular potency. They also exhibited a high probability of gut permeability, and no violation to Lipinski’s rules.

Vismione D showed the highest predictive antifungal and also showed high values as anti-inflammatory agent when compared to other compounds found in EHVG (Table 4). Kaempferol and quercetin showed the highest predictive value as anti-inflammatory. Quercetin and catechin showed the highest values as antioxidant compounds, and anthraquinone, fluconazole, and amphotericin B had no antioxidant predictive activities with Pa values lower than 0.3.

### 2.6. Compounds Present in EHVG, Especially Vismione D, Interact with CaCYP51

The following compounds identified in EHVG were used for molecular docking: anthraquinone F, catechin, kaempferol, and vismione D. Figure 11 illustrates the molecular interaction between vismione D and the enzyme CaCYP51 of *C. albicans*. Analysis of the vismione D + CaCYP51 complex is characterized by the formation of hydrogen bonds between the ligand and the enzyme residues Tyr118, Ser378, and Met508, as well as hydrophobic contact with residues Leu121, Phe228, Pro230, Phe233, Gly303, Ile304, Gly307, Gly308, Thr311, Leu376, Phe380, and Ser508, including interactions with the heme group.

Molecular docking analysis indicated that vismione D presents the highest affinity for the fungal enzyme CaCYP51, with a binding free energy (∆Gbind) of 10.96 kcal/mol and an inhibition constant (Ki) of 0.009 μM, when compared to the reference drugs posaconazole and fluconazole. Anthraquinone provided results similar to those of posaconazole, and the results for catechin and kaempferol were similar to those of fluconazole (Table 5).

Figure 12 shows the spatial conformation obtained for the molecular docking analysis of vismione D and CaCYP51 structures. Several binding sites could be observed, which favor a possible action of the compound on the enzyme.

## 3. Discussion

*V. guianensis* is a native plant found in different regions of Brazil, especially in the Amazon region, where its leaf extracts are used to treat skin inflammatory and infectious skin diseases [13]. Little is known about its mechanisms of action. Thus, our aim was to investigate its anti-*Candida* activity from the hydroethanolic extract prepared with a vegetal part (EHVG), considering that plants are still the main source of bioactive compounds and that there is a growing resistance to commercial antifungal drugs.

For the extract preparation, the particle size of the leaves and the product obtained were evaluated and classified as a semi-fine powder, with a large part retained in mesh M60, followed by mesh M80. It is important to highlight that the particle size of the dry plant material is important for the extract preparation, since the more homogenous the extraction is, the better, resulting in an increased yield and collection of chemical components, maximizing biological activities [19].

The preparation of EHVG with different hydromodules resulted in a similar yield, but different dry and total residues after evaluation of the ash content. The hydromodule 1:10 showed the best result for the chemical analysis, as it exhibited the best qualitative profile due to the more intense peaks and larger and better-defined areas, to confirm the presence of secondary metabolites in the extract. For this reason, it was used for antifungal assays.

Direct flow injection analysis of EHVG with ionization in negative mode confirmed the presence of phenolic compounds, including anthraquinones, catechins, epicatechins, kaempferol, vismione D, and flavonoids. The presence of these compounds was also previously reported by Seo et al. [20], who isolated five benzophenones, vismiaguianones A–E, and two benzocoumarins, vismiaguianins from chloroformic root extract during the investigation of the antitumoral effects of *V. guianensis*. In 2004, Politi et al. [14] described the presence of anthraquinones, vismiones, flavonoids, xanthones, and benzophenones in *n*-hexane, as well as dichloromethane, when investigating leaf methanol extracts in comparison with those from roots. Later, Hussain et al. [21], in a review related to the genus *Vismia*, described the presence of the same compounds which we found, and discussed anthraquinones and vismione as important chemical markers of this group.

According to Costa et al. [22], antifungal activities may be related to the presence of mono- and sesquiterpene metabolites, components also identified in the present study. This previous study demonstrated that such metabolites could promote cell membrane rupture, causing fungal death as well as impairments of membrane synthesis, cellular respiration, and spore germination. Flavonoids may also cause rupture of the microbial membrane due to their lipophilic nature [23]. Other *V. guianensis* chemical components, such as humulene epoxide II (not reported in this study) isolated from the oil of leaves, exhibited antifungal activity against several *Candida* species, including *C. albicans* and *C. glabrata* [22].

EHVG exhibited low toxicity using different techniques and different concentrations. The decision to investigate this by different assays was based on previous observations on the MTT assay; neutral red and crystal violet staining assays provided complementary data for cell viability [24].

The antifungal activity of EHVG was evaluated by time-kill assays, using the *C. albicans* (SC5314) and *C. glabrata* (ATCC 2001) strains. These strains were chosen due to their clinical interest [22,25,26]. EHVG showed a fungicidal effect against both strains, inhibiting all *Candida* strains over 48 h, especially for *C. glabrata* after the first 12 h.

As EHVG was antimicrobial against *C. albicans* and *C. glabrata*, we investigated its effects on *Candida* virulence (adhesion and biofilm formation). The extract impaired the adhesion of all tested strains with similar efficacy to that observed for reference drugs such as fluconazole. This is an important action, considering that adhesion is the first step of a successful fungal colonization and pathogenesis [1,7].

EHVG exerted an inhibitory concentration-dependent effect on early biofilm formation by *C. albicans* (ATCC90028) and *C. glabrata* (ATCC2001). The observed inhibitory effects were equivalent to Amphotericin B, but no differences were observed after incubation of sub-inhibitory concentrations of EHVG. The extract also exerted inhibitory activity on mature biofilms produced by *C. albicans* (ATCC 90028) and *C. glabrata* (ATCC 2001), with similar performance to those of amphotericin B and fluconazole when tested at 2× MIC. For *C. albicans* (SC5314), EHVG showed better results than fluconazole.

The biofilm acts as a protective barrier for the microorganism, rendering their cells more resistant to the immune system, as well as to the effects of antifungal therapies. Biofilms arising from *Candida* infections are difficult to treat and, depending on the biofilm phase, higher concentrations of antifungal compounds are necessary in comparison to planktonic cells [27]. In addition, biofilm formation may vary among *Candida* strains in response to different antifungals, contributing to increased virulence [5,7]. EHVG exhibited anti-*Candida* activity, reducing both adhesion and biofilm formation by all tested strains.

In order to better understanding the underlying mechanisms of EHVG anti-*Candida* activities, molecular modeling was used to identify the probable targets of EHVG compounds. We found that all tested compounds exhibited favorable interactions with CaCYP51 (cytochrome P450 lanosterol 14a-demethylase). However, vismione D was the most potent inhibitor of the enzyme, with a docking score of −10.96kcal/mol. Vismione D also exhibited the highest affinity for CaCYP51, whilst presenting the lowest inhibition constant and high negative binding free energy, with values even greater than those of posaconazole and fluconazole, used herein as reference drugs. These findings indicate vismione D as the most promising anti-*Candida* compound of EHVG. It is important to note that the other tested compounds were also predicted to interact with CaCYP51, allowing us to conclude that EHVG activity may be due to a phytocomplex present in the EHVG, in which vismione D may be the most effective.

Binding free energy is a parameter used to indicate how spontaneously the interaction between a molecule and a biological target occurs. Negative binding free energies indicate that these interactions favor the formation of ligand-receptor complexes [28].

Intermolecular interactions, such as hydrogen bonds with amino acid residues (Tyr-118, Ser-378 and Met-508), hydrophobic interactions, and interactions with the heme group, confer the ability to stabilize a given ligand at the binding site, forming ligand-receptor complexes. In the present study, vismione D was predicted to interact with practically the same amino acids as posaconazole, a commercially available fungicide, but more spontaneously and with a greater interaction potential. In addition, the spatial conformation obtained indicated a more linear structure of vismione D contributing to the molecular interactions.

Values lower than 2Å indicate that the docking protocol is valid, i.e., shows similarity with the experimental structure [28]. The CaCYP51 enzyme plays a key role in the synthesis of ergosterol from its precursor, lanosterol [29,30]. Blockade of this enzyme thus inhibits the process necessary for *Candida* survival [31]. Based on this, it is reasonable to propose that EHVG’s anti-*Candida* activities are due to its ability to affect ergosterol synthesis.

It is important to emphasize the relevance of phytocomplexes present in EHVG for the development of new antifungals with, possibly, a broader spectrum of action than traditional agents [22], since the extract compounds may act additively as antifungals. The in silico absorption, distribution, metabolism, excretion (ADME), and toxicity modeling is an important tool for rational drug design for studying interactions of ligands with biological targets at the atomic scale. As shown by the in silico results, the evaluated compounds, including vismione D, presented good predictive values with high gut absorption, as well as accordance to the Lipinski’s rule, an important characteristic for drug-likeness [32,33].

The in silico absorption, distribution, metabolism, excretion (ADME), and toxicity (T) modeling is an important tool for rational drug design and to study the interactions of ligands with the biological targets in atomic scale, especially if we consider the several restrictions for animal use in the world.

The molecular docking findings suggest that the compounds present in EHVG meet Lipinski’s rules, which indicates that they might act as effective drug candidates with a low risk level for oral use. Furthermore, the ADMET predictor confirmed the in vitro results related to the low toxicity as an important characteristic to show the plant compounds as a good medicinal product [32,33].

Additionally, based on PASS prediction simulation, it is possible to propose other biological activities for vismione D, since it showed a higher value of Pa (potential activity) [34] as an antifungal compound, and good values as an anti-inflammatory and antioxidant compound.

To the best of our knowledge, there are no available data evaluating the in silico activities of compounds from *V. guianensis* extract (EHVG) or any other species of the genus *Vismia*. This highlights the importance of this analysis for elucidating the possible mechanisms underlying the antifungal action of this species.

## 4. Materials and Methods

### 4.1. Collection and Identification of Vismia guianensis

The leaves were collected between October and November 2016, in the morning, in São Luís, MA, Brazil (2°28′47.1″ S and 44°13′17.4″ W). The collected material was cleaned and left to dry for 7 days, protected from sunlight, under natural ventilation, and at room temperature (±24 °C). The plant material was identified and deposited in the Herbarium of Maranhão, Federal University of Maranhão (No. 11.078), and is registered with the National System for the Management of Genetic Heritage and Associated Traditional Knowledge (SisGen) (Registration No. AFE7A08) as regulated by Law No. 13.123/15.

### 4.2. Preparation of the Extract

The dried leaves were ground in an electrical grinder. The powder obtained was used to prepare the extract and the different hydromodules The extract yield was calculated as % of the total biomass of the plant material. The dependence of the extract yield on the hydromodule was studied to determine the best extraction process, and the choice of 1:10 was based on the highest extract yield and best chemical profile.

For the hydromodules, 10 g of the powdered leaves was transferred into a Falcon polyethylene test tube, and added to different Extract/Ethanol ratios (hydromodule) as follows: hydomodule 1:5 (50 mL); hydromodule 1:10 (100 mL); hydromodule 1:15 (150 mL); and 1:20 (200 mL). After this, the extraction process was carried out.

The extract was obtained by maceration for 7 days, protected from light. After this period, the solvent was evaporated in a rotary evaporator, and then the material was lyophilized and stored in a refrigerator for subsequent assessment.

### 4.3. Characterization of the Plant Material

#### 4.3.1. Particle Size Analysis of the Leaves

For characterization of the plant material, two samples of 100 g of dried leaves were placed in a mesh strainer, with mesh sizes No. 16, 20, 40, 60, 80, and 120, and openings of 1190, 850, 425, 250, 180, and 125 μm, respectively, according to Brazilian normalization rules. This lasted for 20 min, with the rheostat being adjusted to intensity 7. The samples were removed and weighed for yield calculation [35].

#### 4.3.2. Ash Content and Extract Yield

Total ash content was quantified as previously described [35]. Samples of the *V. guianensis* leaf powder (3 g), in duplicate, were placed in crucibles, stabilized for 30 min in a desiccator, and heated separately in a muffle furnace (Magnus Ltd.a.^®^) at an initial temperature of 200 °C, with increments of 200 °C every 2 h, until a temperature of 600 °C was reached. The samples were then removed, transferred to a desiccator, maintained at room temperature, and weighed for calculation of the ash volume. At the end of this process, the yield was calculated by placing 1 mL of the extract on Petri dishes and incubated at 37 °C until completely dried. Individual samples were tested in triplicate, and the yield was calculated considering the dry weight of the extract.

### 4.4. Chemical Characterization

#### 4.4.1. Analysis of EHVG by HPLC-PDA

The extract was analyzed by HPLC-PDA. A cleaning step was performed to remove any contaminants. Sample solution (30 mg/mL, 1 mL) was run through solid phase extraction (SPE) using Phenomenex Strata C18 cartridges (500.0 mg stationary phase), which had previously been activated with 5.0 mL MeOH and then equilibrated with 5.0 mL MeOH:H_2_O (1:1, *v*/*v*). The sample was then filtered through a 0.22 μm PTFE filter, and subsequently dried. The dried sample was dissolved at a concentration of 10.0 mg/mL in HPLC-grade methanol solvent. An aliquot of 10.0 μL was injected directly into the HPLC-PDA. The system used was a Shimadzu model HPLC system (Shimadzu Corp., Kyoto, Japan), consisting of a solvent injection module with a binary pump and PDA detector (SPA-20A. The column used was Luna 5.0 μm C18 100 ^0^A (150.0 μm × 4.6 μm). The elution solvents were A (2% acetic acid in water) and B (2% acetic acid in methanol). The sample was eluted according to the following gradient: 5% to 60% B in 60 min. The flow rate was 1 mL/min. The data were processed using the LC Solution software (Shimadzu) [36].

#### 4.4.2. Analysis of EHVG by FIA-ESI-IT-MS^n^

For direct flow injection analysis (FIA-ESI-IT-MS^n^), 10 mg of the crude extract was dissolved in 1 mL of MeOH: H_2_O (1:1, *v*/*v*). After incubation in an ultrasound bath for 5 min, the extract was filtered (0.22 μm), and 20-µL aliquots at a concentration of 5 ppm were directly injected into the FIA-ESI-IT-MS^n^ system.

Full-scan mass spectra were recorded in the range of 100–1000 *m*/*z*. Multistage fragmentations (ESI-MS^n^) were obtained by the collision-induced dissociation (CID) method, using helium for ion activation. The different compounds of EHVG were identified, based on the comparison of UV spectra and characteristic fragmentations, to those from the data in the literature [30].

### 4.5. In Vitro Tests for Anti-Candida Activity

#### 4.5.1. Isolation of Microorganisms

Four *Candida* spp. reference strains (ATCC), including three *C. albicans* strains and one *C. glabrata* strain (ATCC 2001), as well as eight clinical isolates, were used after approval by the Brazilian Ethics Committee (No. 813.402/2014) (Table 6). The strains were obtained from the collection of the Laboratory of Applied Microbiology, Universidade CEUMA, São Luís, MA, Brazil. All *Candida* isolates were identified by multiplex PCR. The identification was performed as previously described [37], using the combination of eight different species-specific primers in a single PCR tube, by combining two yeast-specific fragments from the ITS1 and ITS2 regions and species-specific primers for *C. albicans* and *C. glabrata.* This method was chosen as it allows for the identification of clinical isolates with high specificity, and has the potential to discriminate individual *Candida* species in polyfungal infections. For the experiments, the isolates were subcultured on Sabouraud-dextrose agar (SDA) with chloramphenicol for 24 h at 37 °C.

#### 4.5.2. Inoculum Preparation

The different strains were activated on SDA with chloramphenicol by seeding and incubation for 24 h at 35 °C. The colonies were resuspended in 5 mL of 0.85% sterile saline (0.145 mol/L; 8.5 g/L NaCl). The resulting suspension was homogenized in a Vortex^®^ for 15 s. The cell density was adjusted in a spectrophotometer (Global Trade Technology), at a wavelength of 530 nm, by the addition of saline to obtain an absorbance corresponding to 0.5 McFarland standards.

#### 4.5.3. Determination of Minimum Inhibitory Concentration (MIC)

Antifungal susceptibility tests were standardized following previously established guidelines (M27-A3, [38]), and according to Ostrosky et al. [39]. For the test, 100 µL RPMI 1640 medium (with glutamine and phenol red, no bicarbonate) buffered with MOPS [3-(N-morpholine) propanesulfonic acid] was added to the wells of a 96-well sterile microtiter plate, which contained a cell suspension of *Candida* spp. (1 × 10^6^ CFU/mL). Next, 100 µL EHVG at a concentration of 50 mg/mL was added to the first column, followed by serial dilutions up to 0.095 mg/mL. Amphotericin B (0.0313–16 µg/mL) and Fluconazole (0.125–256 µg/mL) (Sigma-Aldrich, São Paulo, Brazil) were used as positive controls. RPMI 1640 medium, without extract or an antifungal agent, served as the growth control. The *Candida* spp. inoculum plus EHVG or antifungal agent was incubated for 24 h at 37 °C.

The result was analyzed visually and read in a microplate reader at 540 nm. The MIC was defined as the lowest concentration of the extract or antifungal agent at which no growth was visible or detected. The test was carried out in triplicate in two different experiments.

#### 4.5.4. Determination of Minimum Fungicidal Concentration (MFC)

The MFC was determined based on the MIC results. A 10 µL aliquot of the wells corresponding to up to 4× the MIC was seeded onto SDA plates with chloramphenicol. The plates were incubated for 24–48 h at 37 °C, and the MFC was defined as the lowest concentration of the EHVG and antifungal agents that inhibited fungal growth. All tests were carried out in triplicate. The MFC/MIC ratio was calculated in order to determine whether the extract had fungistatic (MFC/MIC ≥ 4) or fungicidal activity (MFC/MIC ≤ 4) [40].

#### 4.5.5. Time-Kill Assay

*C. albicans* ATCC 2001 and *C. albicans* SC 5314 were used to establish the time-kill curves according to [26], with modifications. The inoculum was diluted in RPMI 1640 medium to a final concentration of 5 × 10^3^ CFU/mL, and added to 96-well plates. EHVG was used at concentrations corresponding to the MIC and 2× MIC, using 1% DMSO as vehicle. Amphotericin B plus inoculum was used as a positive control, and RPMI 1640 plus inoculum as a negative control. The plates were incubated for 24 h at 37 °C, and the number of CFUs was determined.

Growth kinetics were evaluated after different periods of incubation: 0, 3, 6, 12, 24, 36, and 48 h. Serial dilutions were prepared for each sample (10^−1^ to 10^−4^), and 10 µL was transferred to Petri dishes containing SDA with chloramphenicol. After colony counting, the following formula was applied: CFU/mL = number of counted colonies × 10^n^/q, where *n* corresponds to the absolute value of the chosen dilution and *q* to the volume (mL) of each dilution seeded on the plates [41]. The assay was carried out in triplicate in two different experiments.

#### 4.5.6. Adhesion Assay

*Candida* spp. was activated as described in item 4.5.2, washed with PBS (3 mL), and centrifuged twice at 2060× *g* for 5 min. After the last wash, the inoculum was resuspended in 5 mL PBS, read at 530 nm, and adjusted to 1 × 10^7^ CFU/mL [42].

For adhesion assessment, 100 µL PBS and 100 µL EHVG, at concentrations corresponding to the MIC and ½ MIC, were added to each *Candida* strain. Serial dilutions were prepared, and the samples were incubated for 90 min at 37 °C. The supernatant was then discarded, and the plates were washed twice with PBS. In the last wash, PBS was discarded completely, and the cells were resuspended in 100 µL PBS. The microdroplet technique was used to seed dilutions of 10^−2^ to 10^−4^ on SDA plates with chloramphenicol, which were incubated for 24–48 h at 37 °C. The assay was carried out in triplicate in two different experiments.

#### 4.5.7. Effect of EHVG on Biofilm Formation

The experiments were carried out as described previously [24], with modifications. Sub-inhibitory concentrations of EHVG (200 µL), corresponding to ¼ MIC and ½ MIC, were used to assess the extract’s interference with biofilm formation. The plates were incubated for 24 h at 37 °C. After this period, the supernatant was removed, and the pellet was washed twice with PBS. The number of CFUs was determined, and the metabolic activity was evaluated by the MTT assay, or by biomass formation using crystal violet staining. For the MTT (3-(4,5-dimethyl-2-thiazyl)-2,5-diphenyl-2H-tetrazolium bromide) assay, a concentration of 5 mg/mL was added, and the samples were read in a microplate reader at 570 nm. For crystal violet staining, 0.2 mg/mL of the dye was added, and the samples were read at 595 nm, as described by Zago et al. [41].

For the quantification of CFU, dilutions of 10^−2^ to 10^−4^ were seeded on SDA plates with chloramphenicol by the microdroplet technique, and the plates were incubated for 24–48 h at 37 °C. For CFU quantification, the formula described in item 2.5.6 was applied. The assay was carried out in triplicate in two different experiments.

The effect of the extract on the mature biofilm was analyzed following the same procedure as described above, and the microplates were incubated for 48 h at 37 °C. The culture medium was changed after 24 h. The supernatant was aspirated, and the biofilm was washed twice with PBS. Concentrations of EHVG (200 µL) corresponding to 2×- and 4× the MIC were used. The plates were incubated at 37 °C for an additional 24 h. After this period, the plates were washed twice with 100 µL PBS, and the number of CFUs, the metabolic activity (MTT assay), and the biomass (crystal violet staining) were analyzed [43] (Seneviratne et al., 2016). For CFU quantification, dilutions of 10^−2^ to 10^−4^ were seeded on SDA plates with chloramphenicol by the microdroplet technique, and the plates were incubated for 24–48 h at 37 °C. The number of CFUs was determined by applying the formula proposed previously [41]. In all experiments, biofilms without extract were used as negative controls, and biofilms with Amphotericin B and Fluconazole were used as positive controls.

### 4.6. Determination of EHVG Cytotoxicity by the MTT Assay

The cytotoxicity of the extracts against murine macrophages (RAW 264.7; APABCAM, Rio de Janeiro, Brazil) was evaluated. The cells were maintained in cell culture flasks (TPP) in RPMI medium supplemented with 10% fetal bovine serum (Invitrogen, New York, NY, USA) and 1% penicillin-streptomycin (Gibco, Grand Island, NE, USA), and then incubated for 1 h at 37 °C in the presence of 5% CO_2_. After reaching sub-confluence, the cells were detached with a cell scraper (TPP) and centrifuged at 2000 rpm for 5 min.

The supernatant was discarded, and the cells were, again, resuspended in RPMI medium and transferred to microtiter plates (100 µL, 1 × 10^6^/well). The plates were incubated for 4 h in the presence of 5% CO_2_ for adherence of the cells. After this period, increasing concentrations of EHVG (0.5 to 50 mg/mL) were added, and the plates were incubated for 24 h at 37 °C in the presence of 5% CO_2_. As growth controls, the cells were incubated with RPMI without extract and 1% DMSO as vehicle.

After incubation, the supernatant was removed, and adherent cells were evaluated by the addition of 10 µL MTT solution (5 mg/mL) (Sigma, St. Louis, MO, USA) to the wells in the dark. The plates were incubated for 4 h at 37 °C. The supernatant was removed, and formazan was extracted by the addition of 100 μL sodium dodecyl sulfate (SDS) and incubation overnight. Absorbance was read at 570 nm after 24 h of incubation.

### 4.7. The Hemolytic Assay

Defibrinated sheep blood (EB FARMA, Rio de Janeiro, Brazil) was used in this study. Red blood cells were isolated by centrifugation at 290× *g* (5810R Centrifuge, Eppendorf) for 10 min at 4 °C. After the removal of plasma, the red blood cells were washed three times with PBS (pH 7.4) and immediately resuspended in 2% (*v*/*v*) of the same buffer. To evaluate the hemolytic activity of the extracts and fractions, 100-µL aliquots of the red blood cell solution were added to flat-bottom 96-well microplates (Kasvi, Italy), together with serial dilutions of the extract and fractions (0.05 to 50 mg/mL). Total hemolysis was achieved with 1% Triton X-100 (Sigma-Aldrich), and PBS was used as a negative control. DMSO (1%, vehicle) was also included as a control. After incubation for 60 min at room temperature, the cells were centrifuged at 300× *g* for 10 min, and absorbance was measured in the supernatant at 540 nm [11]. Relative hemolytic activity was expressed in relation to Triton X-100, and was calculated using the following formula: relative hemolytic activity (%) = [(As − Ab) × 100]/(Ac − Ab), where Ac is the absorbance of the control (blank, without extract), As is the absorbance in the presence of the extract, and Ab is the absorbance in the presence of Triton X-100. The assays were carried out in quadruplicate on three different experiments.

### 4.8. Evaluation of Cell Viability by the Neutral Red Assay

Neutral red (Vetec/Sigma Aldrich, São Paulo, Brazil) was prepared in PBS (20 µg/mL), and 100 µL/well of this solution was placed in the wells that contained RAW cells (100 µL/1 × 10^6^) treated with different concentrations of EHVG, starting at 50 mg/mL (25, 12.5, 6.25, 3.125, 1.76, 0.78, and 0.39 mg/mL). After 2 h of incubation while protected from light, the plates were centrifuged, and the supernatant was discarded. Next, ethanol P.A. (100 µL) was added to the wells. The plates were shaken on an orbital shaker for 15 min. Absorbance was then determined at 570 nm, and the values obtained were converted into percentage of cell viability [44].

### 4.9. In Silico Analysis

#### 4.9.1. Ligands and Target Preparations

The GaussView 5.0.8 program was used to illustrate the three-dimensional (3D) structure of the compounds Vismione D, anthraquinone, kaempferol, and catechin, identified by HPLC-UV/Vis and FIA-ESI-IT-MS in the crude extract of *V. guianensis* leaves. The geometrical and vibrational properties were calculated in vacuum by the density functional theory combined with the 6-311++G** (d, p) basis set, using Gaussian 09. The 3D structure of 14-alpha-demethylase from *C. albicans* (CaCYP51) was obtained from the Protein Data Bank (PDB) (# 5FSA), then resolved by X-ray crystallography with a resolution of 2.86 Å.

#### 4.9.2. Molecular Docking

The structures of CaCPY51 and of the ligands were prepared for the molecular docking calculations using the AutoDock Tools (ADT), version 1.5.6. The structure of CaCYP51 was considered rigid, while each ligand was considered flexible. Gasteiger partial charges were calculated after the addition of all hydrogens. Next, the apolar hydrogens of CaCYP51 and of the ligands were subsequently fused. A cubic box of 120 × 120 × 120 points, with spacing of 0.35 Å, was generated throughout the portion that corresponded to the active site of the macromolecule.

The grid box was centered on the heme group of CaCPY51. For molecular docking, a global search was performed using Lamarckian Genetic Algorithms (AGL), and a local search (LS) using pseudo-Solis and Wets. Each ligand was submitted to 100 independent simulations of ligand binding, and the remaining fit parameters were set to default values. The initial coordinates of the interactions between CaCYP51 and the compounds present in EHVG were chosen using the criterion of clustering low-energy conformations, combined with visual inspection [37]. Molecular docking was performed using AutoDock 4.2.

#### 4.9.3. Pharmacokinetics and Toxicity Measurement

The SwissADME online method was used to assess the pharmacokinetic properties (ADME) of the compounds. Lipinski’s five rule was determined based on the following parameters: molecular weight not more than 500 Dalton; H-bond donors ≤ 5; H-bond acceptors ≤ 10; molar refractivity ranging from 40 to 130; and lipophilicity positive drug-like properties of any compound [45]. In addition, the online tool admetSAR (http://lmmd.ecust.edu.cn/admetsar2, accessed on 28 September 2022) was used to calculate the toxicological properties of all the compounds.

#### 4.9.4. Pass Prediction

The PASS online tool (http://www.pharmaexpert.ru/passonline/predict.php, accessed on 28 September 2022) was used to determine the other potential biological activities of the selected compounds. The potential biological activity considered the Pa and Pi values to range from 0.000 to 1.000 [46,47], and Pa values higher than the Pi values were considered to have the following characteristics: Pa < 0.7 suggests high drug activities; 0.5 < Pa < 0.7 shows moderate therapeutic potentials; and Pa < 0.5 shows poor pharmaceutical activity [48,49].

### 4.10. Statistical Analysis

The results were expressed as the mean ± standard deviation. One-way analysis of variance (ANOVA) was used, followed by Dunn’s post-test for multiple comparisons and the Student *t*-test for comparison between two groups. All analyses were performed using the GraphPad Prism 8.0 software. The significance adopted was 5% (*p* < 0.05).

## 5. Conclusions

EHVG has important anti-*Candida* activity, being able to attenuate the growth and virulence factors of *Candida* ATCC strains and clinical isolates of interest. This anti-*Candida* effect may be due to the presence of vismione D, which may act synergistically to other compounds. This inhibits the Candida CaCYP51 enzyme, and, therefore, impairs the biosynthesis of ergosterol.

Overall, the results suggest that EHVG is a promising anti-*Candida* alternative that can be used as a reference to generate a lead compound for the purpose of antifungal drug discovery.

## Figures and Tables

**Figure 1 antibiotics-11-01834-f001:**
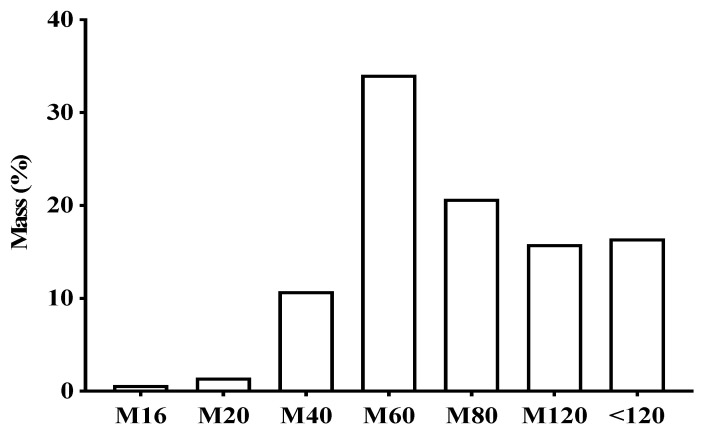
Percentage of different particles sizes in the powder of dry *Vismia guianensis* leaves after grinding and sifting using different mesh sizes.

**Figure 2 antibiotics-11-01834-f002:**
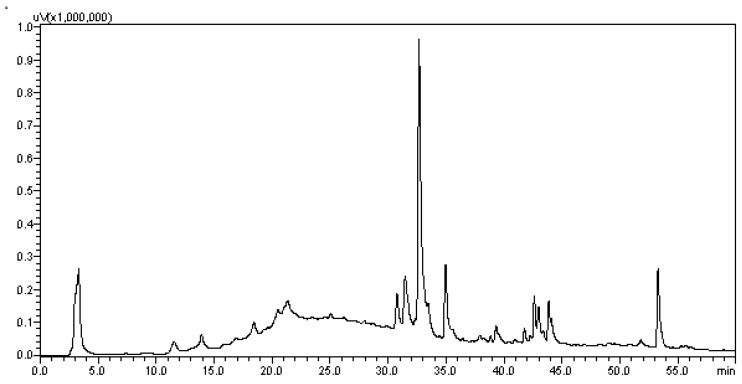
Chromatogram of the hydroalcoholic extracts of *Vismia guianensis* obtained by HPLC with UV detection (HPLC–PDA) for the hydromodule 1:10 with defined peaks and areas.

**Figure 3 antibiotics-11-01834-f003:**
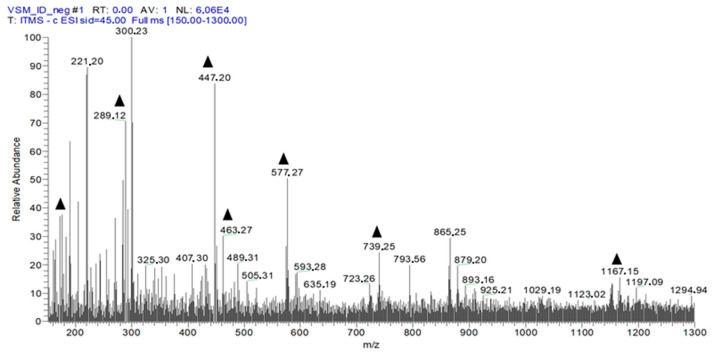
First-order spectrum of direct flow injection analysis (FIA-ESI-IT-MS) obtained in the negative mode for EHVG. (▲) Fragments of the chemical compounds identified.

**Figure 4 antibiotics-11-01834-f004:**
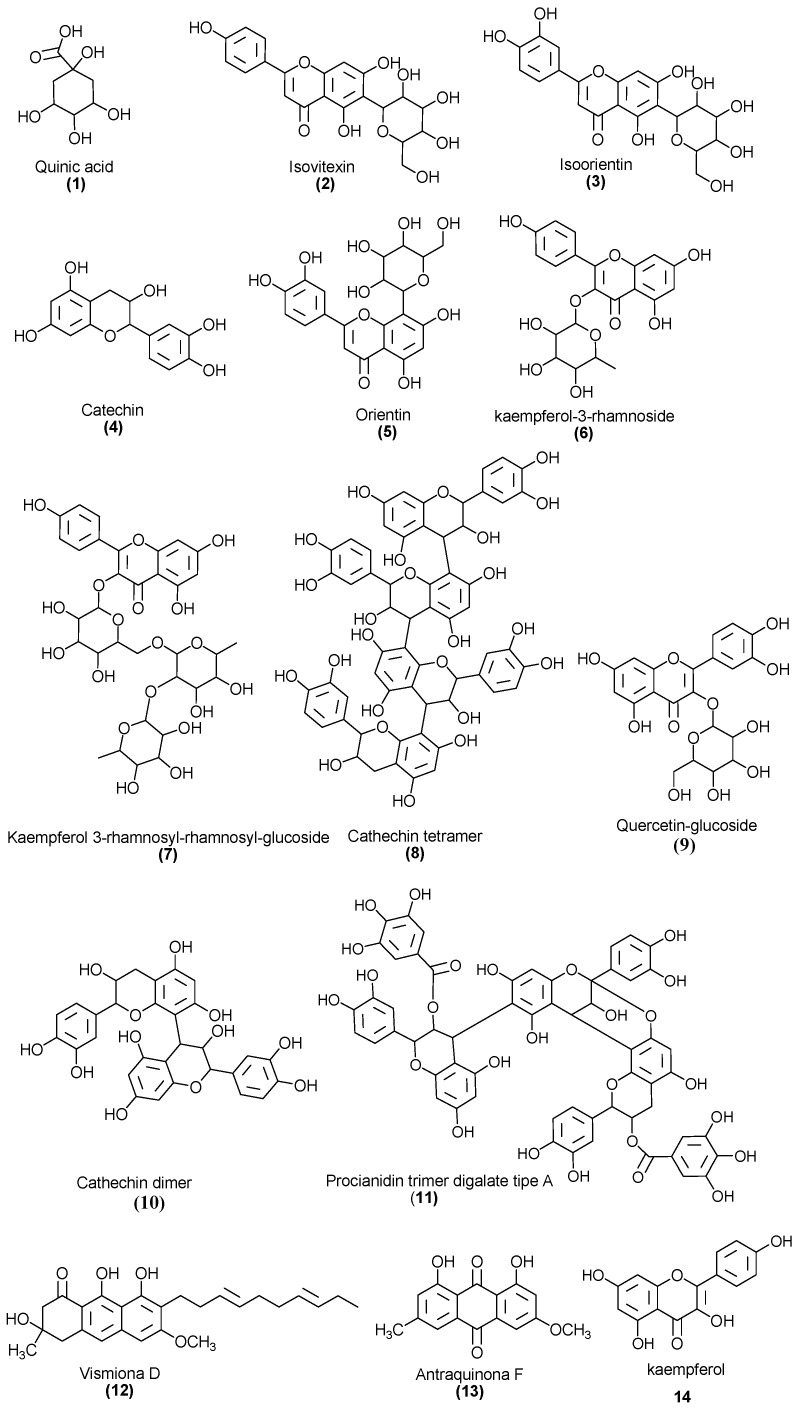
Chemical structures of the compounds identified in the *Vismia guianensis* leaf extract.

**Figure 5 antibiotics-11-01834-f005:**
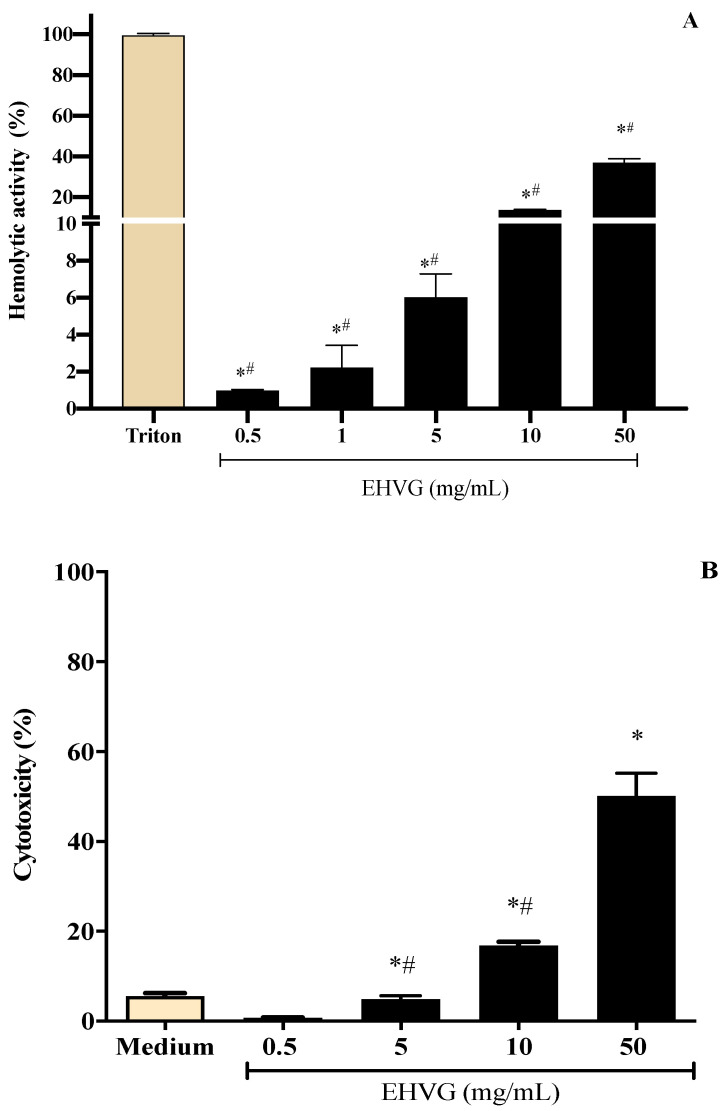
Different concentrations of the hydroalcoholic leaf extract of *Vismia guianensis* (EHVG) showed low toxicity when evaluated by the hemolysis (**A**) and MTT assay (**B**). Triton was the positive control for hemolysis (**A**) and medium was the negative control in the cytotoxic assay. Data represent the mean ± standard deviation of individual samples tested in quadruplicate. (*) *p* < 0.05 compared to the control; (#) *p* < 0.05 compared with other concentrations.

**Figure 6 antibiotics-11-01834-f006:**
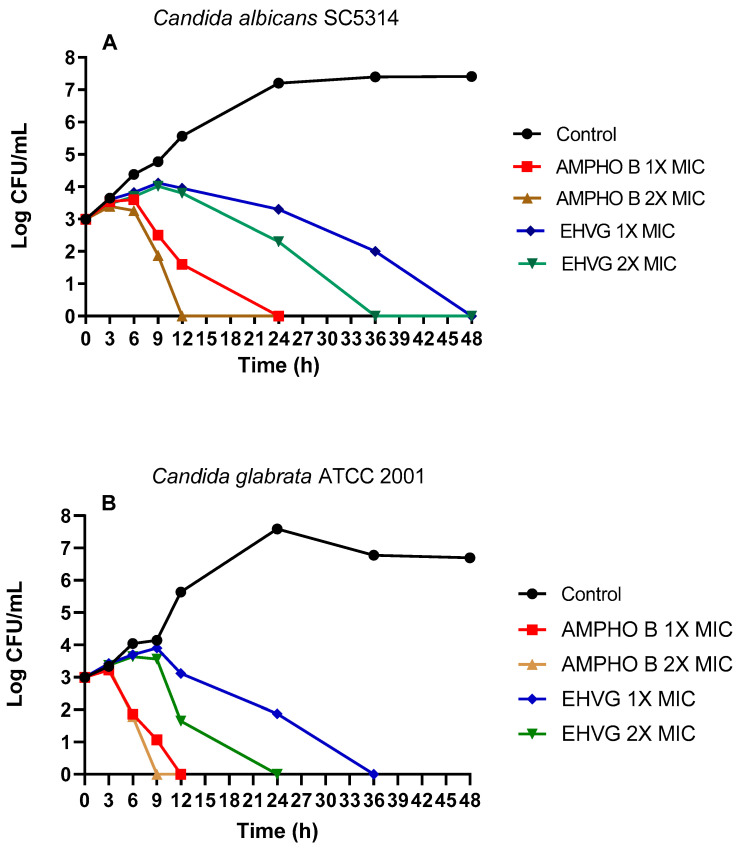
Effects of EHVG on time-kill curves. The effects of EHVG against *Candida albicans* SC5314 (**A**) and *Candida glabrata* ATCC 2001 (**B**) were tested at two different concentrations corresponding to the MIC or 2× MIC, then compared to untreated cultures (Control) or with cultures treated with Amphotericin B (AMPHO B; MIC—0.5 µg/mL and 2× MIC—1 µg/mL).

**Figure 7 antibiotics-11-01834-f007:**
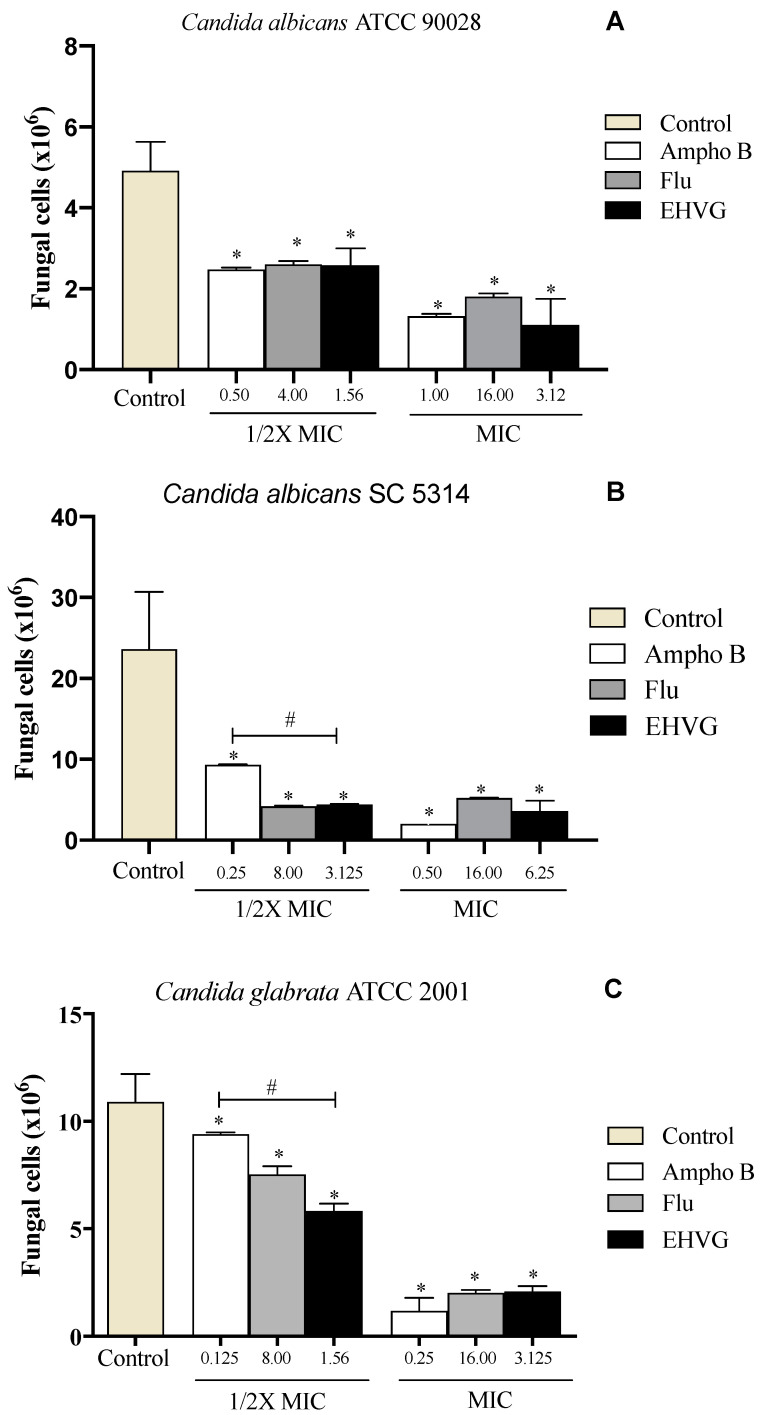
EHVG inhibited the adhesion of *Candida albicans* ATCC 90,028 (**A**), *C. albicans* SC5314 (**B**), and *C. glabrata* ATCC 2001 (**C**) in cultures treated with concentrations of ½ MIC and MIC when compared to untreated samples (control). The EHVG-treated cultures were also compared to those treated with the reference drugs Amphotericin B (Ampho B) or Fluconazole (Flu). (*) *p* < 0.05 differs from untreated control; (#) *p* < 0.05 differs from Amphotericin B.

**Figure 8 antibiotics-11-01834-f008:**
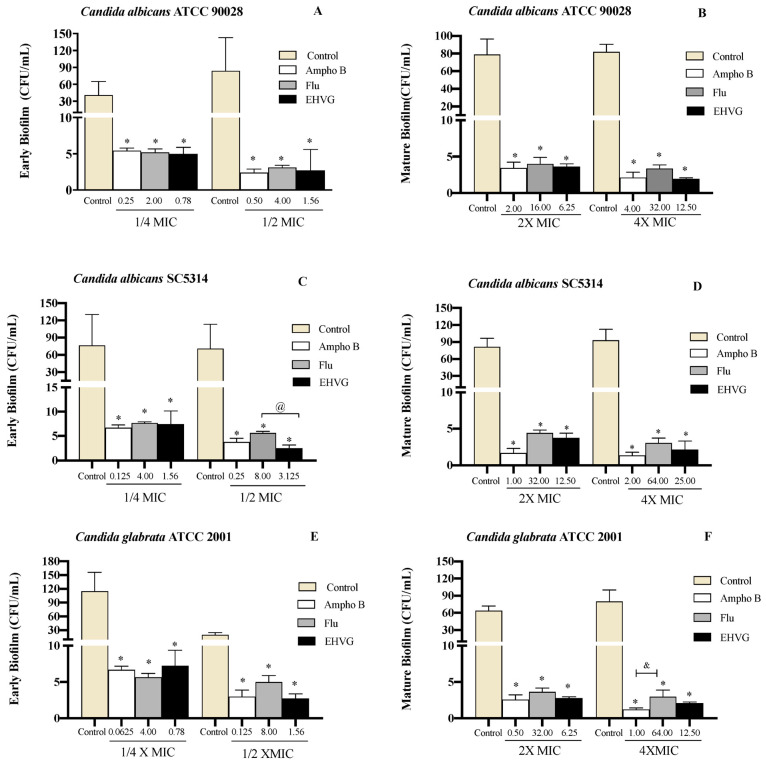
EHVG inhibited the formation of early (**A**,**C**,**E**) and mature (**B**,**D**,**F**) biofilms. The CFU/mL values were established in cultures of *Candida albicans* (ATCC 90028, **A**,**B**; SC5314, **C**,**D**) and *C. glabrata* (ATCC 2001, **E**,**F**) after 48 h and 72 h. The samples treated with different concentrations of EHVG (1/4 and 1/2× MIC for early biofilms, and 2× and 4× MIC for mature biofilms) were compared to cultures treated with Amphotericin B (Ampho B) or fluconazole (Flu), and with the untreated control. (*) *p* < 0.05 in comparison to control; (@) in comparison to Fluconazole, and (&) *p* < 0.05 Fluconazole compared to Amphotericin B.

**Figure 9 antibiotics-11-01834-f009:**
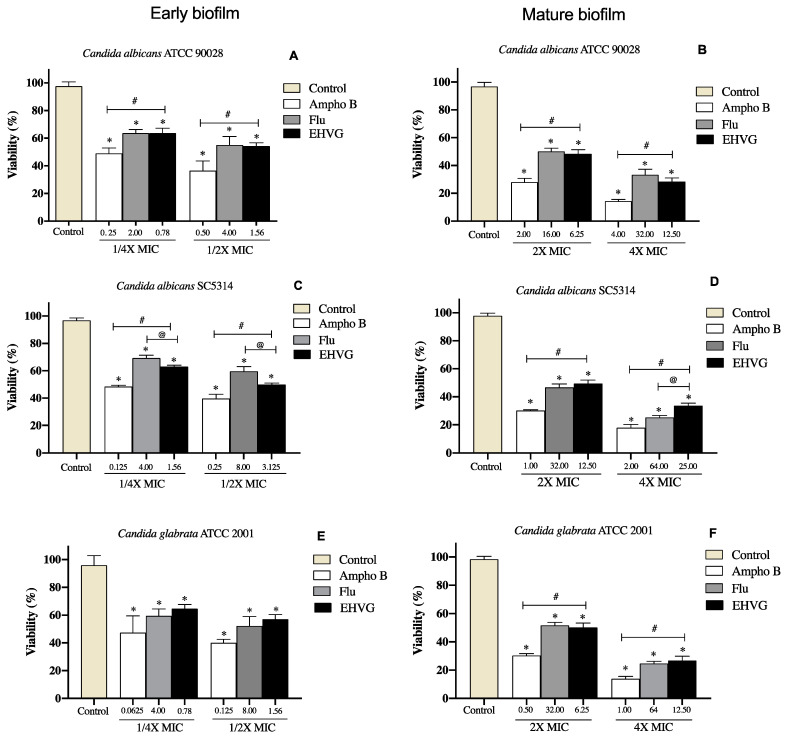
Percentage of inhibition on EHVG-treated early (**A**,**C**,**E**) and mature biofilms (**B**,**D**,**F**) of *Candida albicans* ATCC 90,028 (**A**,**B**), *C. albicans* SC5314 (**C**,**D**) and *C. glabrata* ATCC 2001 (**E**,**F**) after 48 and 72 h, evaluated by MTT assay. Samples were treated with EHVG (1/4 and 1/2 for early biofilms, and 2× and 4× MIC for mature biofilms) and compared to untreated controls or the reference drugs Amphotericin B (Ampho B) and Fluconazole (Flu). (*) *p* < 0.05 compared to control. (#) *p* < 0.05 compared to Amphotericin B and (@) *p* <0.05 compared to fluconazole.

**Figure 10 antibiotics-11-01834-f010:**
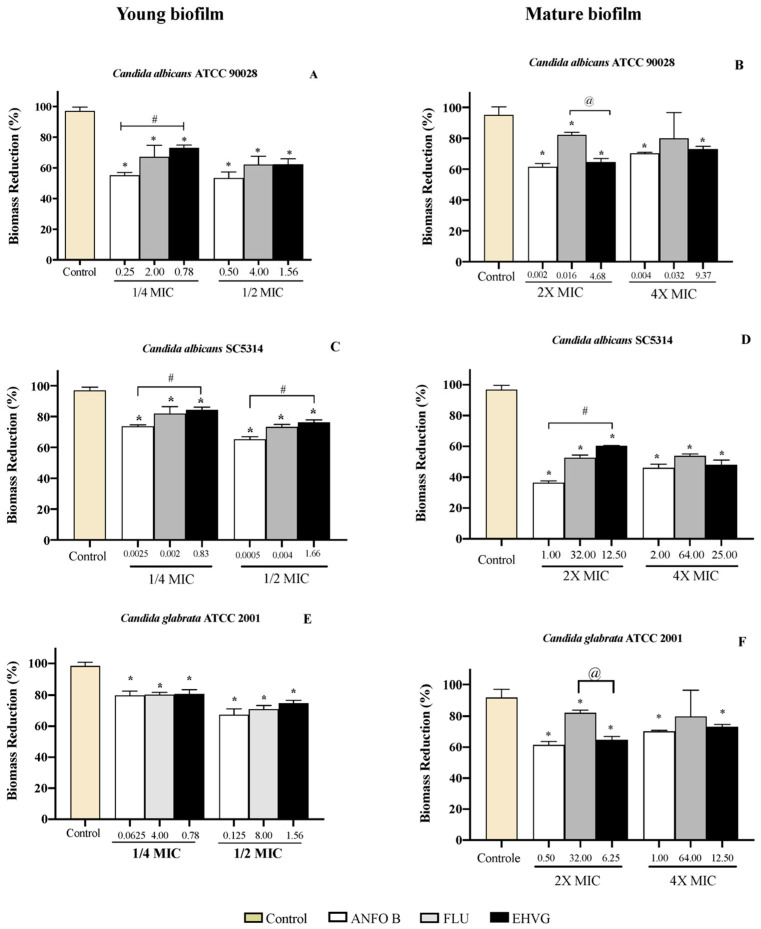
Reductions in biofilm biomass stained with crystal violet (CV) of *Candida* spp. (ATCC 90028, **A**,**B**), (ATCC 2001, **C**,**D**) and (SC5314, **E**,**F**) after 48 and 72 h. The samples were treated with EHVG and compared to untreated controls or to the reference drugs Amphotericin B (AMHO B) and Fluconazole (FLU). (*) *p* < 0.05 differs from control; (#) *p* < 0.05 differs from Amphotericin B and (@) *p* <0.05 compared to fluconazole.

**Figure 11 antibiotics-11-01834-f011:**
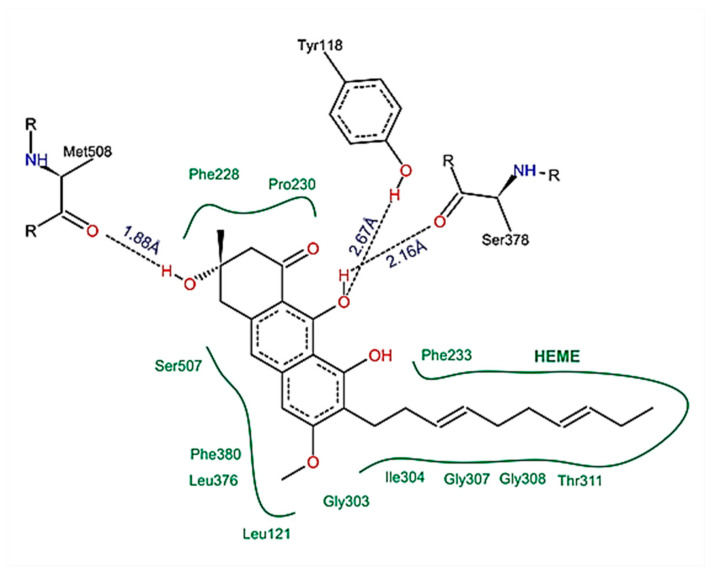
Schematic representation of the interactions between vismione D and CaCYP51, identified by molecular docking. It is possible to observe the formation of hydrogen bonds and hydrophobic contacts between the ligand and the enzyme residues, including interactions with the heme group. The figure was obtained with PoseView.

**Figure 12 antibiotics-11-01834-f012:**
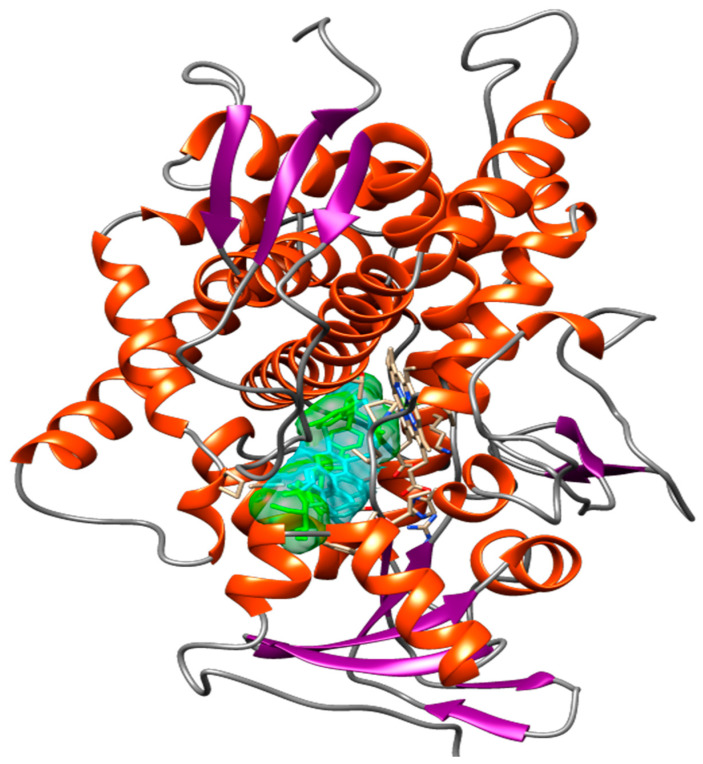
Spatial conformation showing several common binding sites between the CaCYP51 enzyme (PDB: 5FSA) and Vismione D (blue) or the antifungal posaconazole (green) through molecular docking. The image was obtained with USCF Chimera.

**Table 1 antibiotics-11-01834-t001:** Yield of the different hydromodules of the hydroalcoholic leaf extract of *Vismia guianensis*.

Hydromodule	Dry Residue (X ± SD) ^a^	Total Residue	Extract Yield (%)
1:5	21 ± 0.0018	210	11
1:10	13 ± 0.0004	258	13
1:15	8 ± 0.0006	119	12
1:20	7 ± 0.0017	15	13

^a^: X ± SD = mean ± standard deviation.

**Table 2 antibiotics-11-01834-t002:** Compounds identified by MS^n^ in *Vismia guianensis* extract.

Number	[M^−^H]^−^	MS^n^ Ions	Proposed Compound
**1**	191	173, 111, 85	Quinic acid
**2**	431	269	Isovitexin
**3**	447	429, 357	Isoorientin
**4**	289		Catechin
**5**	447	429, 301, 269, 229	Orientin
**6**	431	285, 163	Kaempferol-*O*-rhamnoside
**7**	731	285, 255	Kaempferol galactoside-rhamnoside
**8**	1153	1001, 983, 789	Catechin tetramer
**9**	463	301, 283, 273, 229, 179, 121	Quercetin glycoside
**10**	577	425, 407, 285, 257, 213	Catechin dimer
**11**	1167	1015, 863, 711	A-type procyanidin trimer
**12**	409	273, 255	Vismione D
**13**	283	269, 239	Anthraquinone F
**14**	285		Kaempferol

**Table 3 antibiotics-11-01834-t003:** Minimum inhibitory (MIC) and minimum fungicidal (MFC) concentrations of the hydroalcoholic leaf extract of *Vismia guianensis* (EHVG) against *Candida* spp. strains.

*Candida* Strain	*Vismia guianensis* (EHVG)			Antifungal	
	MIC ^a^	MFC ^a^	MFC/MIC Ratio	Ampho B ^b^	Flu ^c^
*C. glabrata* (ATCC 2001) ^d^	3.125	6.25	2	0.25	16
*C. albicans* (ATCC 90028)	3.125	3.125	1	1	8
*C. albicans* (ATCC 14053)	6.25	6.25	1	0.5	8
*C. albicans* (SC 5314)	6.25	6.25	1	0.5	16
A1 ^e^ *C. albicans*	6.25	12.5	2	1	8
A2 *C. albicans*	3.125	3.125	1	0.5	8
A3 *C. albicans*	6.25	6.25	1	0.25	4
A4 *C. albicans*	3.125	6.25	2	0.5	16
A5 *C. albicans*	3.125	3.125	1	0.5	16
A6 *C. albicans*	3.125	3.125	1	0.5	16
A7 *C. albicans*	6.25	6.25	1	1	16

^(a)^ Values are expressed as mg/mL. ^(b)^ Ampho B = Amphotericin B concentration: 0.03 to 16 µg/mL. ^(c)^ Flu = Fluconazole concentration: 0.125 to 256 µg/mL. ^(d)^ ATCC^®^ (American Type Culture Collection). ^(e)^ A1–A7: clinical samples.

**Table 4 antibiotics-11-01834-t004:** PASS prediction of commercial drugs and three selected compounds detected in the *Vismia guianensis* extract, considering the Potential activity (Pa) higher than potential inactivity (Pi), as well as the antifungal, antioxidant, and anti-inflammatory activities.

Activity/ Compounds	Antifungal		Anti-Inflammatory		Antioxidant	
	Pa ^a^	Pi ^b^	Pa	Pi	Pa	Pi
Vismione D	0.684	0.011	0.606	0.030	0.478	0.008
Catechin	0.552	0.023	0.548	0.044	0.810	0.003
Kaempferol	0.495	0.031	0.676	0.019	0.856	0.003
Quercetin	0.490	0.032	0.689	0.017	0.872	0.003
Anthraquinone	0.351	0.063	0.410	0.090	-	-
Fluconazole	0.726	0.008	-	-	-	-
Amphotericin	0.977	0.000	0.330	0.136	-	-

^a^: Pa (probability “to be active”) estimates the chance that the studied compound belongs to the sub-class of active compounds. ^b^: Pi (probability “to be inactive”) estimates the chance that the studied compound belongs to the sub-class of inactive compounds.

**Table 5 antibiotics-11-01834-t005:** Free binding energies and inhibition constants between fourteen compounds identified in *Vismia guianensis* extract and CaCYP51, obtained through molecular docking.

Ligand	ΔGbind (kcal/mol) *	Ki (μM) **
Vismione D	−10.96	0.009
Anthraquinone F	−7.92	1.56
Catechin	−6.97	7.79
Kaempferol	−6.70	12.35
Quercetin	−5.60	20.98
Posaconazole	−8.43	0.57
Fluconazole	−6.89	11.61

(*) Bind free energy; (**) inhibition constant.

**Table 6 antibiotics-11-01834-t006:** Reference strains tested.

Identification	Type of Strain
ATCC 2001 ^a^—C*. glabrata*	Reference
ATCC 90,028—*C. albicans*	Reference
ATCC 14,053—*C. albicans*	Reference
ATCC MYA 2876 (SC 5314)—*C. albicans*	Reference–Wild type
A1 *C. albicans*	Clinical isolate–vagina
A2 *C. albicans*	Clinical isolate–vagina
A3 *C. albicans*	Clinical isolate–vagina
A4 *C. albicans*	Clinical isolate–vagina
A5 *C. albicans*	Clinical isolate–vagina
A6 *C. albicans*	Clinical isolate–vagina
A7 *C. albicans*	Clinical isolate–oral
A8 *C. albicans*	Clinical isolate–oral

^a^: ATCC—American Type Culture Collection.

## Data Availability

On demand.

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
