# Peer review of "The Anti-Virulence Effect of Vismia guianensis against Candida albicans and Candida glabrata"

_antibiotics, 2022, doi:10.3390/antibiotics11121834_

Round 1

Reviewer 1 Report

The manuscript is focused on the anti-virulence effect of Vismia guianensis against Candida 2 albicans and Candida glabrata. The present manuscript has been written well but I have some major queries?

In the result sub section 3.1.2. Chemical profile of EHVG; Author performed chemical profile of EHVG using HPLC-UV/VIS at 254 nm. hromatographic profile of hydromodule 1:10 that quantitatively and qualitatively exhibited the best extraction profile in terms of peak area. Provide the quantitative data of hydromodule. Peaks does not define the constituents.

The fourteen compounds identified by flow injection analysis (FIA-ESI-IT-MS) are shown in Table 2. It was possible to observe a predominance of anthraquinones, flavo-noids, and Vismione D.

On what basis these 14 compounds identified? Whereas HPLC chromatogram shows only 8 peaks. Justify?

The PASS online tool evaluated the antifungal, anti-inflammatory, and antioxidant activities of the three selected compounds detected in EHVG as Kaempferol, Anthraquinone and Vismione D in comparison with Fluconazole and Amphotericin.

Why author choose only these three constituents for in silico activity? No quantitative data reported by author for the constituents present in the extract.

Author provide the HPLC chromatograms of all hydromodule. Also provide the details and method of phytoconstituents identification?

Why author not used GC-MS analysis for the identification of phytoconstituents present in different hydromodule of EHVG? Provide GC-MS analysis of EHVG hydromodule.

Author prepare extract using ethanol: water (7:3) but it is not clear in methodology section how author prepared different hydromodule and not mention the percentage yield of the maceration extract.

Author mention percentage yield of different hydromodule and no details available for the preparation of preparation of different hydromodule. On the other hand in the particle size analysis mention about the yield calculation.

Preparation of extracts should be more describe.

Author did not provide results of ash content but in material and method section ash content available?

HPLC Analysis: Describe more about the HPLC analysis and provide full gradient system for 60 minutes.

Conclusion should be more emphasize on phytoconstituents present in EHVG and their activity.

Author Response

Reviewer 1

Comments and Suggestions for Authors

The manuscript is focused on the anti-virulence effect of Vismia guianensis against Candida albicans and Candida glabrata. The present manuscript has been written well but I have some major queries.

In the result sub section 3.1.2. Chemical profile of EHVG; Author performed chemical profile of EHVG using HPLC-UV/VIS at 254 nm. chromatographic profile of hydromodule 1:10 that quantitatively and qualitatively exhibited the best extraction profile in terms of peak area. Provide the quantitative data of hydromodule. Peaks does not define the constituents.

Answer: We thank the reviewer for this comment. The quantitative analysis we performed was only comparing peak areas. However, we understand that this is not the ideal method for quantification. Thus, we have now made clear in the manuscript that the analysis was qualitative.

The fourteen compounds identified by flow injection analysis (FIA-ESI-IT-MS) are shown in Table 2. It is possible to observe a predominance of anthraquinones, flavonoids, and vismione D. On what basis these 14 compounds were identified? Whereas HPLC chromatogram shows only 8 peaks. Justify?

Answer: The compounds were identified by FIA-ESI-IT/MS (direct infusion mass spectrometry). Mass spectrometry is a more sensitive detection method than PDA. Thus, some less abundant compounds do not appear in HPLC-PDA but appear in mass spectrometry. The fragments of each molecular ion were compared with the literature to identify the compounds.

The PASS online tool evaluated the antifungal, anti-inflammatory, and antioxidant activities of the three selected compounds detected in EHVG as Kaempferol, Anthraquinone, and Vismione D, in comparison with Fluconazole and Amphotericin. Why author choose only these three constituents for in silico activity? No quantitative data reported by author for the constituents present in the extract.

Answer: We included five compounds considering the antifungal activity, gut permeability, and no violation of Lipinskirules. This information was added to the material and methods and discussion to clarify.

Authors provided the HPLC chromatograms of all hydromodules. Also provide the details and method of phytoconstituents identification.

Answer: We identified the chemical constituents only in the 1:10 hydromodule. This hydromodule was selected for chemical characterization based on the qualitative data obtained from the HPLC-PDA. The identification method is described in section 4.4.2. "The different compounds in EHVG were identified based on comparison of UV spectra, characteristic fragmentations, and literature data."

Why didn´t the authors used GC-MS analysis for the identification of phytoconstituents present in different hydromodule of EHVG? Provide GC-MS analysis of EHVG hydromodule.

Answer: We appreciate the reviewer's suggestion, but the extract has no apolar molecules. The solvent used for extraction was ethanol: water. The compounds are more polar. Therefore, the extract is not suitable for GC-MS analysis. In the case of a derivatization, the same compounds will be identified.

The authors prepared the extract using ethanol: water (7:3) but it is not clear in the methodology section how they prepared different hydromodules and there was no mention of the yield percentage of the macerated extract. The authors mentioned the yield of different hydromodules but no details were available for their preparation. On the other hand in the particle size analysis mention about the yield calculation. Preparation of extracts should be more descriptive.

Answer: Thank you for your observation. We included the information in the material and methods to improve the text.

The extract was prepared using ethanol 70% (7:3). For the hydromodule preparation, 10g powdered V.guianensis leaves were added to variable volumes of ethanol 70% as follows:

Hydromodule 1:5= 50 mL ethanol 70%

Hydromodule 1:10 = 100 mL ethanol 70%

Hydromodule 1:15 = 150 mL ethanol 70%

Hydromodule 1:20 = 200 mL ethanol 70%

Author did not provide results of ash content but in material and method section ash content available?

Answer: The result for total ash content is described in table 1, and the methodology is described in section 4.3.2

HPLC Analysis: Describe more about the HPLC analysis and provide a full gradient system for 60 minutes.

Answer: We thank you for your comment. The change has been added to the manuscript.

Reviewer 2 Report

The manuscript by Motta et al presents an interesting and complete study regarding the determination of the antifungal action of extracts of a plant endemic to the Amazon, Vismia guianensis. The study is well conducted and explores the applicability of a plant that is not very studied, and therefore the study is highly valuable. However, the manuscript still needs a lot of work, mainly on the language and style. I listed below some of the aspects that need improvement:

The abstract needs editing (language and typing).

What does hydromodule mean? I do not understand. What do you mean by 1:10 hydromodule? A dilution 1:10 in water? You have to clarify this.

Please provide details regarding species identification with multiplex PCR (primers, PCR program...)

I noticed that in the assays determining antimicrobial susceptibility (MIC) the authors do not mention the vehicle (1% DMSO?). Please clarify.

Please define EHVG the first time you mentioned (line 100).

The results section would benefit of a more fluid structure, with  text providing linkage between the sections. Providing only a description of each table and figure is limiting for the reader. This comment is valid for sections regarding the production and characterization of the extract. The sections focusing on microbiology are much more clearer.

I do not understand Table 1. Dry residue vs Total residue - is this %? Weight?

line 108: "Figure 2 shows the chromatographic profile of hydromodule 1:10 that quantitatively and qualitatively exhibited the best extraction profile in terms of peak area". The authors have to expand this statement. I do not see anywhere the expected distribution (%) of each compound as determined by the chromatogram.

Please revise numbering of the sections.

Italicize all the species names throughout the manuscript.

line 320: "The presence of these compounds has been previously reported by others". Please expand this idea. On the same extracts? Of the same species? The distribution of the main components varies between extractions? These notions are important to support the relevance of your findings.

I do not understand the meaning of the sentence beginning on line 331 ("The decision...")

Altough not mandatory, since the manuscript is so rich in results, I missed a general conclusion integrating all results.

Author Response

Reviewer 2:

line 108: “Figure 2 shows the chromatographic profile of hydromodule 1:10 that quantitatively and qualitatively exhibited the best extraction profile in terms of peak area”. The authors have to expand this statement. I do not see anywhere the expected distribution (%) of each compound as determined by the chromatogram.

Answer: We thank the reviewer for this comment. The quantitative analysis we performed was only comparing peak areas. However, we understand that this is not the ideal method for quantification. Thus, we have clarified in the manuscript that the analysis was qualitative.

The abstract needs editing (language and typing).

Answer: We thank you for the comment. The abstract and the role manuscript were reviewed by a professional expert.

What does hydromodule mean? I do not understand. What do you mean by 1:10 hydromodule? A dilution 1:10 in water? You have to clarify this.

Answer: Thank you for your observation. We included the information in the material and methods to improve the text.

The extract was prepared using ethanol 70% (7:3). For the hydromodule preparation, 10g powdered V.guianensis leaves were added to variable volumes of ethanol 70% as follows:

Hydromodule 1:5= 50 mL ethanol 70%

Hydromodule 1:10 = 100 mL ethanol 70%

Hydromodule 1:15 = 150 mL ethanol 70%

Hydromodule 1:20 = 200 mL ethanol 70%

Please provide details regarding species identification with multiplex PCR (primers, PCR program...)

Answer: The identification was performed as previously described, using the combination of eight different species-specific primers in a single PCR tube by combining two yeast-specific fragments from the ITS1 and ITS2 regions and species-specific primers for  C. albicans and  C. glabrata.  We choose this method since it allows us to identify clinical isolates with high specificity directly from clinical specimens and can potentially discriminate individual Candida species in polyfungal infections.

I noticed that in the assays determining antimicrobial susceptibility (MIC), the authors do not mention the vehicle (1% DMSO?). Please clarify.

Answer: The DMSO was used for the extract preparation, but the extract was lyophilized and suspended in PBS for the assays. For this reason, we didn’t include DMSO as control.

Please define EHVG the first time you mentioned (line 100).

Answer: Suggestion accepted, and we added the name of the extract to the text

The results section would benefit of a more fluid structure, with text providing linkage between the sections. Providing only a description of each table and figure is limiting for the reader. This comment is valid for sections regarding the production and characterization of the extract. The sections focusing on microbiology are much more clearer.

Answer: We thank you for the comment; this suggestion was partially accepted. It is a difficult decision because sometimes the referees ask to move this additional information for the discussion.

2.7. I do not understand Table 1. Dry residue vs Total residue - is this %? Weight?

Answer: Yes, we will clarify the description of the material and methods

2.8. line 108: "Figure 2 shows the chromatographic profile of hydromodule 1:10 that quantitatively and qualitatively exhibited the best extraction profile in terms of peak area". The authors have to expand this statement. I do not see anywhere the expected distribution (%) of each compound as determined by the chromatogram.

Answer: We thank you for the comment. The quantitative analysis we performed was only comparing peak areas. Still, we understand this is not the ideal quantification method, so we have removed the term “quantitative” from the manuscript.

2.9. Please revise numbering of the sections.

Answer: We thank you for the comment, and sections numbers were reviewed

2.10. Italicize all the species names throughout the manuscript.

Answer: We thank you for the comment, and the names of all species are now Italicized

line 320: "The presence of these compounds has been previously reported by others". Please expand this idea. On the same extracts? Of the same species? The distribution of the main components varies between extractions? These notions are important to support the relevance of your findings.

Answer: We thank you for the comment and included more information related to this issue.

I do not understand the meaning of the sentence beginning on line 331 ("The decision...")

Answer:  We didn’t find this; probably something related to the configuration

Although not mandatory, since the manuscript is so rich in results, I missed a general conclusion integrating all results.

Answer: The general conclusion is item 5 and was placed after material and methods at the end of the manuscript.

Round 2

Reviewer 1 Report

No more suggestion